# Interaction Networks for Learning about Objects, Relations and Physics

## Abstract

Reasoning about objects, relations, and physics is central to human intelligence, and a key goal of artificial intelligence. Here we introduce the *interaction network*, a model which can reason about how objects in complex systems interact, supporting dynamical predictions, as well as inferences about the abstract properties of the system. Our model takes graphs as input, performs object- and relation-centric reasoning in a way that is analogous to a simulation, and is implemented using deep neural networks. We evaluate its ability to reason about several challenging physical domains: n-body problems, rigid-body collision, and non-rigid dynamics. Our results show it can be trained to accurately simulate the physical trajectories of dozens of objects over thousands of time steps, estimate abstract quantities such as energy, and generalize automatically to systems with different numbers and configurations of objects and relations. Our interaction network implementation is the first general-purpose, learnable physics engine, and a powerful general framework for reasoning about object and relations in a wide variety of complex real-world domains.

## 1 Introduction

Representing and reasoning about objects, relations and physics is a "core" domain of human common sense knowledge [25], and among the most basic and important aspects of intelligence [27, 15]. Many everyday problems, such as predicting what will happen next in physical environments or inferring underlying properties of complex scenes, are challenging because their elements can be composed in combinatorially many possible arrangements. People can nevertheless solve such problems by decomposing the scenario into distinct objects and relations, and reasoning about the consequences of their interactions and dynamics. Here we introduce the *interaction network* – a model that can perform an analogous form of reasoning about objects and relations in complex systems.

Interaction networks combine three powerful approaches: structured models, simulation, and deep learning. Structured models [7] can exploit rich, explicit knowledge of relations among objects, independent of the objects themselves, which supports general-purpose reasoning across diverse contexts. Simulation is an effective method for approximating dynamical systems, predicting how the elements in a complex system are influenced by interactions with one another, and by the dynamics of the system. Deep learning [23, 16] couples generic architectures with efficient optimization algorithms to provide highly scalable learning and inference in challenging real-world settings.

Interaction networks explicitly separate how they reason about relations from how they reason about objects, assigning each task to distinct models which are: fundamentally object- and relation-centric; and independent of the observation modality and task specification (see Model section 2 below and Fig. 1a). This lets interaction networks automatically generalize their learning across variable numbers of arbitrarily ordered objects and relations, and also recompose their knowledge of entities

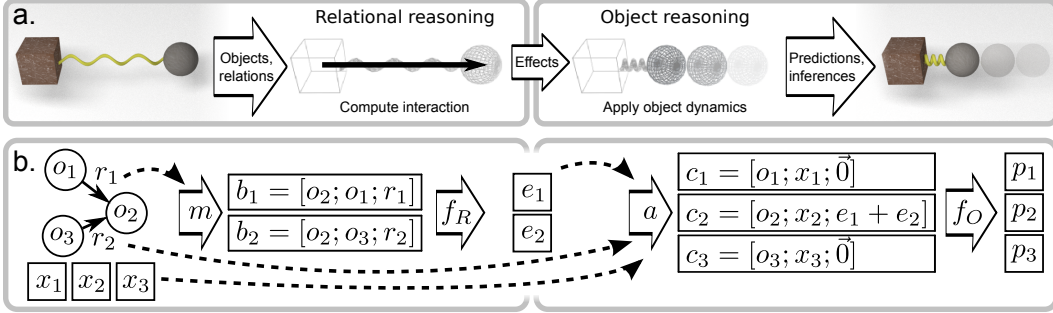

Figure 1: Schematic of an interaction network. *a.* For physical reasoning, the model takes objects and relations as input, reasons about their interactions, and applies the effects and physical dynamics to predict new states. *b.* For more complex systems, the model takes as input a graph that represents a system of objects, $o_j$, and relations, $\langle i, j, r_k \rangle_k$, instantiates the pairwise interaction terms, $b_k$, and computes their effects, $e_k$, via a relational model, $f_R(\cdot)$. The $e_k$ are then aggregated and combined with the $o_j$ and external effects, $x_j$, to generate input (as $c_j$), for an object model, $f_O(\cdot)$, which predicts how the interactions and dynamics influence the objects, $p$.

and interactions in novel and combinatorially many ways. They take relations as explicit input, allowing them to selectively process different potential interactions for different input data, rather than being forced to consider every possible interaction or those imposed by a fixed architecture.

We evaluate interaction networks by testing their ability to make predictions and inferences about various physical systems, including n-body problems, and rigid-body collision, and non-rigid dynamics. Our interaction networks learn to capture the complex interactions that can be used to predict future states and abstract physical properties, such as energy. We show that they can roll out thousands of realistic future state predictions, even when trained only on single-step predictions. We also explore how they generalize to novel systems with different numbers and configurations of elements. Though they are not restricted to physical reasoning, the interaction networks used here represent the first general-purpose learnable physics engine, and even have the potential to learn novel physical systems for which no physics engines currently exist.

**Related work**   Our model draws inspiration from previous work that reasons about graphs and relations using neural networks. The "graph neural network" [22] is a framework that shares learning across nodes and edges, the "recursive autoencoder" [24] adapts its processing architecture to exploit an input parse tree, the "neural programmer-interpreter" [21] is a composable neural network that mimics the execution trace of a program, and the "spatial transformer" [11] learns to dynamically modify network connectivity to capture certain types of interactions. Others have explored deep learning of logical and arithmetic relations [26], and relations suitable for visual question-answering [1].

The behavior of our model is similar in spirit to a physical simulation engine [2], which generates sequences of states by repeatedly applying rules that approximate the effects of physical interactions and dynamics on objects over time. The interaction rules are relation-centric, operating on two or more objects that are interacting, and the dynamics rules are object-centric, operating on individual objects and the aggregated effects of the interactions they participate in.

Previous AI work on physical reasoning explored commonsense knowledge, qualitative representations, and simulation techniques for approximating physical prediction and inference [28, 9, 6]. The "NeuroAnimator" [8] was perhaps the first quantitative approach to learning physical dynamics, by training neural networks to predict and control the state of articulated bodies. Ladický et al. [14] recently used regression forests to learn fluid dynamics. Recent advances in convolutional neural networks (CNNs) have led to efforts that learn to predict coarse-grained physical dynamics from images [19, 17, 18]. Notably, Fragkiadaki et al. [5] used CNNs to predict and control a moving ball from an image centered at its coordinates. Mottaghi et al. [20] trained CNNs to predict the 3D trajectory of an object after an external impulse is applied. Wu et al. [29] used CNNs to parse objects from images, which were then input to a physics engine that supported prediction and inference.

## 2 Model

**Definition** To describe our model, we use physical reasoning as an example (Fig. 1a), and build from a simple model to the full interaction network (abbreviated IN). To predict the dynamics of a single object, one might use an object-centric function, $f_O$, which inputs the object's state, $o_t$, at time $t$, and outputs a future state, $o_{t+1}$. If two or more objects are governed by the same dynamics, $f_O$ could be applied to each, independently, to predict their respective future states. But if the objects interact with one another, then $f_O$ is insufficient because it does not capture their relationship. Assuming two objects and one directed relationship, e.g., a fixed object attached by a spring to a freely moving mass, the first (the *sender*, $o_1$) influences the second (the *receiver*, $o_2$) via their interaction. The effect of this interaction, $e_{t+1}$, can be predicted by a relation-centric function, $f_R$. The $f_R$ takes as input $o_1$, $o_2$, as well as attributes of their relationship, $r$, e.g., the spring constant. The $f_O$ is modified so it can input both $e_{t+1}$ and the receiver's current state, $o_{2,t}$, enabling the interaction to influence its future state, $o_{2,t+1}$,

$$e_{t+1} = f_R(o_{1,t}, o_{2,t}, r) \qquad\qquad o_{2,t+1} = f_O(o_{2,t}, e_{t+1})$$

The above formulation can be expanded to larger and more complex systems by representing them as a graph, $G = \langle O, R \rangle$, where the nodes, $O$, correspond to the objects, and the edges, $R$, to the relations (see Fig. 1b). We assume an attributed, directed multigraph because the relations have attributes, and there can be multiple distinct relations between two objects (e.g., rigid and magnetic interactions). For a system with $N_O$ objects and $N_R$ relations, the inputs to the IN are,

$$O = \{o_j\}_{j=1...N_O} \ , \ R = \{\langle i, j, r_k \rangle_k\}_{k=1...N_R} \text{ where } i \neq j, 1 \leq i, j \leq N_O \ , \ X = \{x_j\}_{j=1...N_O}$$

The $O$ represents the states of each object. The triplet, $\langle i, j, r_k \rangle_k$, represents the $k$-th relation in the system, from sender, $o_i$, to receiver, $o_j$, with relation attribute, $r_k$. The $X$ represents external effects, such as active control inputs or gravitational acceleration, which we define as not being part of the system, and which are applied to each object separately.

The basic IN is defined as,

$$\text{IN}(G) = \phi_O(a(G, \ X, \ \phi_R(\ m(G)\ )\ )) \tag{1}$$

$$
\begin{aligned}
m(G) \quad &= B \ = \ \{b_k\}_{k=1...N_R} & a(G, X, E) \quad &= C \ = \ \{c_j\}_{j=1...N_O} \\
f_R(b_k) \quad &= e_k & f_O(c_j) \quad &= p_j \\
\phi_R(B) \quad &= E \ = \ \{e_k\}_{k=1...N_R} & \phi_O(C) \quad &= P \ = \ \{p_j\}_{j=1...N_O}
\end{aligned} \tag{2}
$$

The marshalling function, $m$, rearranges the objects and relations into interaction terms, $b_k = \langle o_i, o_j, r_k \rangle \in B$, one per relation, which correspond to each interaction's receiver, sender, and relation attributes. The relational model, $\phi_R$, predicts the effect of each interaction, $e_k \in E$, by applying $f_R$ to each $b_k$. The aggregation function, $a$, collects all effects, $e_k \in E$, that apply to each receiver object, merges them, and combines them with $O$ and $X$ to form a set of object model inputs, $c_j \in C$, one per object. The object model, $\phi_O$, predicts how the interactions and dynamics influence the objects by applying $f_O$ to each $c_j$, and returning the results, $p_j \in P$. This basic IN can predict the evolution of states in a dynamical system – for physical simulation, $P$ may equal the future states of the objects, $O_{t+1}$.

The IN can also be augmented with an additional component to make abstract inferences about the system. The $p_j \in P$, rather than serving as output, can be combined by another aggregation function, $g$, and input to an abstraction model, $\phi_A$, which returns a single output, $q$, for the whole system. We explore this variant in our final experiments that use the IN to predict potential energy.

An IN applies the same $f_R$ and $f_O$ to every $b_k$ and $c_j$, respectively, which makes their relational and object reasoning able to handle variable numbers of arbitrarily ordered objects and relations. But one additional constraint must be satisfied to maintain this: the $a$ function must be commutative and associative over the objects and relations. Using summation within $a$ to merge the elements of $E$ into $C$ satisfies this, but division would not.

Here we focus on binary relations, which means there is one interaction term per relation, but another option is to have the interactions correspond to $n$-th order relations by combining $n$ senders in each $b_k$. The interactions could even have variable order, where each $b_k$ includes all sender objects that interact with a receiver, but would require a $f_R$ than can handle variable-length inputs. These possibilities are beyond the scope of this work, but are interesting future directions.

119  **Implementation**  The general definition of the IN in the previous section is agnostic to the choice
120  of functions and algorithms, but we now outline a learnable implementation capable of reasoning
121  about complex systems with nonlinear relations and dynamics. We use standard deep neural network
122  building blocks, multilayer perceptrons (MLP), matrix operations, etc., which can be trained efficiently
123  from data using gradient-based optimization, such as stochastic gradient descent.

124  We define $O$ as a $D_S \times N_O$ matrix, whose columns correspond to the objects' $D_S$-length state vectors.
125  The relations are a triplet, $R = \langle R_r, R_s, R_a \rangle$, where $R_r$ and $R_s$ are $N_O \times N_R$ binary matrices which
126  index the receiver and sender objects, respectively, and $R_a$ is a $D_R \times N_R$ matrix whose $D_R$-length
127  columns represent the $N_R$ relations' attributes. The $j$-th column of $R_r$ is a one-hot vector which
128  indicates the receiver object's index; $R_s$ indicates the sender similarly. For the graph in Fig. 1b,
129  $R_r = \begin{bmatrix} 0 & 0 \\ 1 & 1 \\ 0 & 0 \end{bmatrix}$ and $R_s = \begin{bmatrix} 1 & 0 \\ 0 & 0 \\ 0 & 1 \end{bmatrix}$. The $X$ is a $D_X \times N_O$ matrix, whose columns are $D_X$-length vectors
130  that represent the external effect applied each of the $N_O$ objects.

131  The marshalling function, $m$, computes the matrix products, $OR_r$ and $OR_s$, and concatenates them
132  with $R_a$:    $m(G) = [OR_r; OR_s; R_a] = B$ .
133  The resulting $B$ is a $(2D_S + D_R) \times N_R$ matrix, whose columns represent the interaction terms, $b_k$,
134  for the $N_R$ relations (we denote vertical and horizontal matrix concatenation with a semicolon and
135  comma, respectively). The way $m$ constructs interaction terms can be modified, as described in our
136  Experiments section (3).

137  The $B$ is input to $\phi_R$, which applies $f_R$, an MLP, to each column. The output of $f_R$ is a $D_E$-length
138  vector, $e_k$, a distributed representation of the effects. The $\phi_R$ concatenates the $N_R$ effects to form the
139  $D_E \times N_R$ effect matrix, $E$.

140  The $G$, $X$, and $E$ are input to $a$, which computes the $D_E \times N_O$ matrix product, $\bar{E} = ER_r^T$, whose
141  $j$-th column is equivalent to the elementwise sum across all $e_k$ whose corresponding relation has
142  receiver object, $j$. The $\bar{E}$ is concatenated with $O$ and $X$:    $a(G, X, E) = [O; X; \bar{E}] = C$.
143  The resulting $C$ is a $(D_S + D_X + D_E) \times N_O$ matrix, whose $N_O$ columns represent the object states,
144  external effects, and per-object aggregate interaction effects.

145  The $C$ is input to $\phi_O$, which applies $f_O$, another MLP, to each of the $N_O$ columns. The output of $f_O$
146  is a $D_P$-length vector, $p_j$, and $\phi_O$ concatenates them to form the output matrix, $P$.

147  To infer abstract properties of a system, an additional $\phi_A$ is appended and takes $P$ as input. The $g$
148  aggregation function performs an elementwise sum across the columns of $P$ to return a $D_P$-length
149  vector, $\bar{P}$. The $\bar{P}$ is input to $\phi_A$, another MLP, which returns a $D_A$-length vector, $q$, that represents
150  an abstract, global property of the system.

151  Training an IN requires optimizing an objective function over the learnable parameters of $\phi_R$ and $\phi_O$.
152  Note, $m$ and $a$ involve matrix operations that do not contain learnable parameters.

153  Because $\phi_R$ and $\phi_O$ are shared across all relations and objects, respectively, training them is statisti-
154  cally efficient. This is similar to CNNs, which are very efficient due to their weight-sharing scheme.
155  A CNN treats a local neighborhood of pixels as related, interacting entities: each pixel is effectively
156  a receiver object and its neighboring pixels are senders. The convolution operator is analogous to
157  $\phi_R$, where $f_R$ is the local linear/nonlinear kernel applied to each neighborhood. Skip connections,
158  recently popularized by residual networks, are loosely analogous to how the IN inputs $O$ to both
159  $\phi_R$ and $\phi_O$, though in CNNs relation- and object-centric reasoning are not delineated. But because
160  CNNs exploit local interactions in a fixed way which is well-suited to the specific topology of images,
161  capturing longer-range dependencies requires either broad, insensitive convolution kernels, or deep
162  stacks of layers, in order to implement sufficiently large receptive fields. The IN avoids this restriction
163  by being able to process arbitrary neighborhoods that are explicitly specified by the $R$ input.

## 3  Experiments

165  **Physical reasoning tasks**  Our experiments explored two types of physical reasoning tasks: pre-
166  dicting future states of a system, and estimating their abstract properties, specifically potential energy.
167  We evaluated the IN's ability to learn to make these judgments in three complex physical domains:
168  n-body systems; balls bouncing in a box; and strings composed of springs that collide with rigid
169  objects. We simulated the 2D trajectories of the elements of these systems with a physics engine, and
170  recorded their sequences of states. See the Supplementary Material for full details.

In the n-body domain, such as solar systems, all $n$ bodies exert distance- and mass-dependent gravitational forces on each other, so there were $n(n-1)$ relations input to our model. Across simulations, the objects' masses varied, while all other fixed attributes were held constant. The training scenes always included 6 bodies, and for testing we used 3, 6, and 12 bodies. In half of the systems, bodies were initialized with velocities that would cause stable orbits, if not for the interactions with other objects; the other half had random velocities.

In the bouncing balls domain, moving balls could collide with each other and with static walls. The walls were represented as objects whose shape attribute represented a rectangle, and whose inverse-mass was 0. The relations input to the model were between the $n$ objects (which included the walls), for $(n(n-1)$ relations). Collisions are more difficult to simulate than gravitational forces, and the data distribution was much more challenging: each ball participated in a collision on less than 1% of the steps, following straight-line motion at all other times. The model thus had to learn that despite there being a rigid relation between two objects, they only had meaningful collision interactions when they were in contact. We also varied more of the object attributes – shape, scale and mass (as before) – as well as the coefficient of restitution, which was a relation attribute. Training scenes contained 6 balls inside a box with 4 variably sized walls, and test scenes contained either 3, 6, or 9 balls.

The string domain used two types of relations (indicated in $r_k$), relation structures that were more sparse and specific than all-to-all, as well as variable external effects. Each scene contained a string, comprised of masses connected by springs, and a static, rigid circle positioned below the string. The $n$ masses had spring relations with their immediate neighbors ($2(n-1)$), and all masses had rigid relations with the rigid object ($2n$). Gravitational acceleration, with a magnitude that was varied across simulation runs, was applied so that the string always fell, usually colliding with the static object. The gravitational acceleration was an external input (not to be confused with the gravitational attraction relations in the n-body experiments). Each training scene contained a string with 15 point masses, and test scenes contained either 5, 15, or 30 mass strings. In training, one of the point masses at the end of the string, chosen at random, was always held static, as if pinned to the wall, while the other masses were free to move. In the test conditions, we also included strings that had both ends pinned, and no ends pinned, to evaluate generalization.

Our model takes as input the state of each system, $G$, decomposed into the objects, $O$ (e.g., n-body objects, balls, walls, points masses that represented string elements), and their physical relations, $R$ (e.g., gravitational attraction, collisions, springs), as well as the external effects, $X$ (e.g., gravitational acceleration). Each object state, $o_j$, could be further divided into a dynamic state component (e.g., position and velocity) and a static attribute component (e.g., mass, size, shape). The relation attributes, $R_a$, represented quantities such as the coefficient of restitution, and spring constant. The input represented the system at the current time. The prediction experiment's target outputs were the velocities of the objects on the subsequent time step, and the energy estimation experiment's targets were the potential energies of the system on the current time step. We also generated multi-step rollouts for the prediction experiments (Fig. 2), to assess the model's effectiveness at creating visually realistic simulations. The output velocity, $v_t$, on time step $t$ became the input velocity on $t + 1$, and the position at $t + 1$ was updated by the predicted velocity at $t$.

**Data**  Each of the training, validation, test data sets were generated by simulating 2000 scenes over 1000 time steps, and randomly sampling 1 million, 200k, and 200k one-step input/target pairs, respectively. The model was trained for 2000 epochs, randomly shuffling the data indices between each. We used mini-batches of 100, and balanced their data distributions so the targets had similar per-element statistics. The performance reported in the Results was measured on held-out test data.

We explored adding a small amount of Gaussian noise to 20% of the data's input positions and velocities during the initial phase of training, which was reduced to 0% from epochs 50 to 250. The noise std. dev. was $0.05\times$ the std. dev. of each element's values across the dataset. It allowed the model to experience physically impossible states which could not have been generated by the physics engine, and learn to project them back to nearby, possible states. Our error measure did not reflect clear differences with or without noise, but rollouts from models trained with noise were slightly more visually realistic, and static objects were less subject to drift over many steps.

**Model architecture**  The $f_R$ and $f_O$ MLPs contained multiple hidden layers of linear transforms plus biases, followed by rectified linear units (ReLUs), and an output layer that was a linear transform plus bias. The best model architecture was selected by a grid search over layer sizes and depths. All

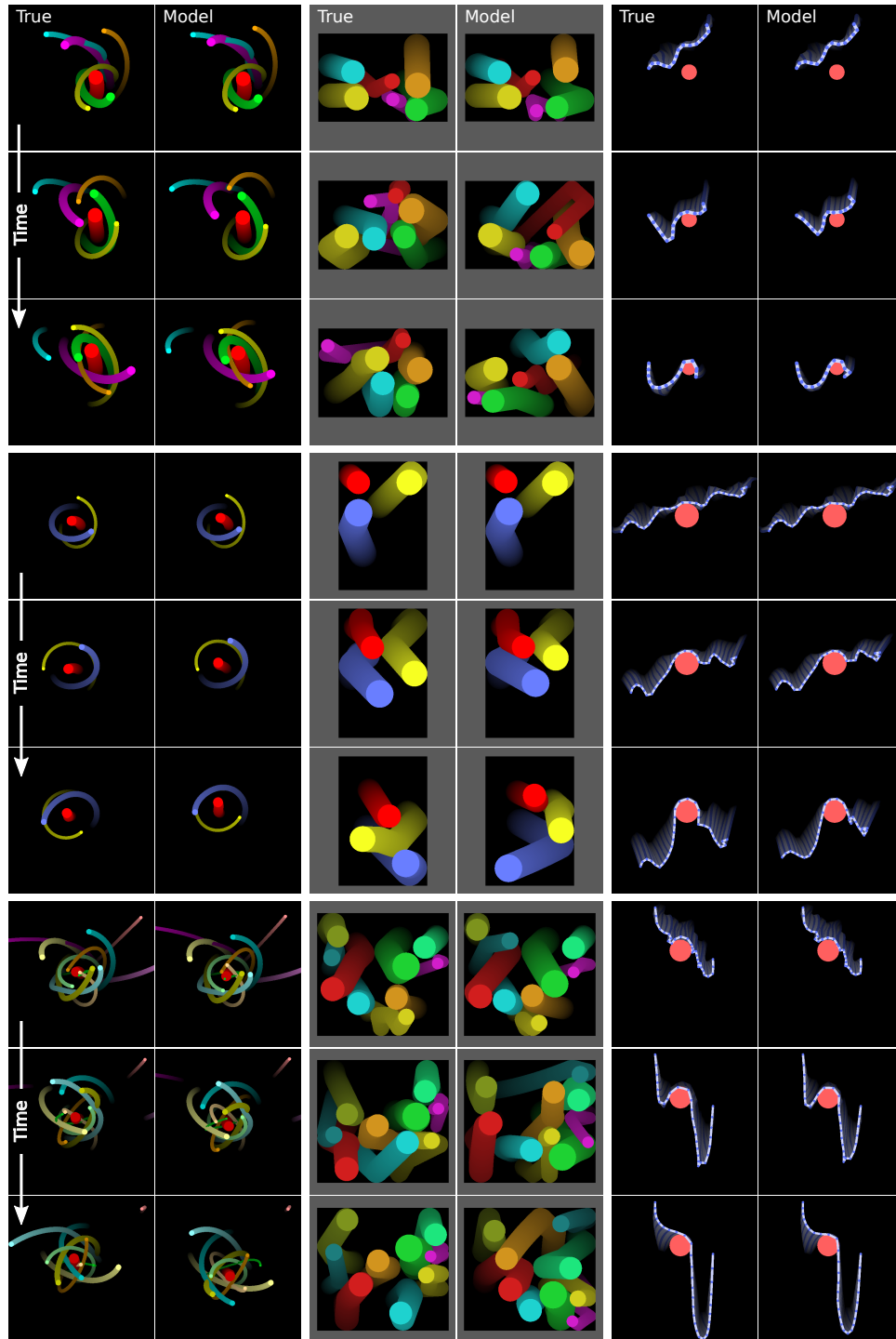

Figure 2: Prediction rollouts. Each column contains three panels of three video frames (with motion blur), each spanning 1000 rollout steps. Columns 1-2 are ground truth and model predictions for n-body systems, 3-4 are bouncing balls, and 5-6 are strings. Each model column was generated by a single model, trained on the underlying states of a system of the size in the top panel. The middle and bottom panels show its generalization to systems of different sizes and structure. For n-body, the training was on 6 bodies, and generalization was to 3 and 12 bodies. For balls, the training was on 6 balls, and generalization was to 3 and 9 balls. For strings, the training was on 15 masses with 1 end pinned, and generalization was to 30 masses with 0 and 2 ends pinned.

inputs (except $R_r$ and $R_s$) were normalized by centering at the median and rescaling the 5th and 95th percentiles to -1 and 1. All training objectives and test measures used mean squared error (MSE) between the model's prediction and the ground truth target.

All prediction experiments used the same architecture, with parameters selected by a hyperparameter search. The $f_R$ MLP had four, 150-length hidden layers, and output length $D_E = 50$. The $f_O$ MLP had one, 100-length hidden layer, and output length $D_P = 2$, which targeted the $x, y$-velocity. The $m$ and $a$ were customized so that the model was invariant to the absolute positions of objects in the scene. The $m$ concatenated three terms for each $b_k$: the difference vector between the dynamic states of the receiver and sender, the concatenated receiver and sender attribute vectors, and the relation attribute vector. The $a$ only outputs the velocities, not the positions, for input to $\phi_O$.

The energy estimation experiments used the IN from the prediction experiments with an additional $\phi_A$ MLP which had one, 25-length hidden layer. Its $P$ inputs' columns were length $D_P = 10$, and its output length was $D_A = 1$.

We optimized the parameters using Adam [13], with a waterfall schedule that began with a learning rate of 0.001 and down-scaled the learning rate by 0.8 each time the validation error, estimated over a window of 40 epochs, stopped decreasing.

Two forms of L2 regularization were explored: one applied to the effects, $E$, and another to the model parameters. Regularizing $E$ improved generalization to different numbers of objects and reduced drift over many rollout steps. It likely incentivizes sparser communication between the $\phi_R$ and $\phi_O$, prompting them to operate more independently. Regularizing the parameters generally improved performance and reduced overfitting. Both penalty factors were selected by a grid search.

Few competing models are available in the literature to compare our model against, but we considered several alternatives: a constant velocity baseline which output the input velocity; an MLP baseline, with two 300-length hidden layers, which took as input a flattened vector of all of the input data; and a variant of the IN with the $\phi_R$ component removed (the interaction effects, $E$, was set to a 0-matrix).

# 4   Results

**Prediction experiments**    Our results show that the IN can predict the next-step dynamics of our task domains very accurately after training, with orders of magnitude lower test error than the alternative models (Fig. 3a, d and g, and Table 1). Because the dynamics of each domain depended crucially on interactions among objects, the IN was able to learn to exploit these relationships for its predictions. The dynamics-only IN had no mechanism for processing interactions, and performed similarly to the constant velocity model. The baseline MLP's connectivity makes it possible, in principle, for it to learn the interactions, but that would require learning how to use the relation indices to selectively process the interactions. It would also not benefit from sharing its learning across relations and objects, instead being forced to approximate the interactive dynamics in parallel for each objects.

The IN also generalized well to systems with fewer and greater numbers of objects (Figs. 3b-c, e-f and h-k, and Table SM1 in Supp. Mat.). For each domain, we selected the best IN model from the system size on which it was trained, and evaluated its MSE on a different system size. When tested on smaller n-body and spring systems from those on which it was trained, its performance actually exceeded a model trained on the smaller system. This may be due to the model's ability to exploit its greater experience with how objects and relations behave, available in the more complex system.

We also found that the IN trained on single-step predictions can be used to simulate trajectories over thousands of steps very effectively, often tracking the ground truth closely, especially in the n-body and string domains. When rendered into images and videos, the model-generated trajectories are usually visually indistinguishable from those of the ground truth physics engine (Fig. 2; see Supp. Mat. for videos of all images). This is not to say that given the same initial conditions, they cohere perfectly: the dynamics are highly nonlinear and imperceptible prediction errors by the model can rapidly lead to large differences in the systems' states. But the incoherent rollouts do not violate people's expectations, and might be roughly on par with people's understanding of these domains.

**Estimating abstract properties**    We trained an abstract-estimation variant of our model to predict potential energies in the n-body and string domains (the ball domain's potential energies were always 0), and found it was much more accurate (n-body MSE 1.4, string MSE 1.1) than the MLP baseline

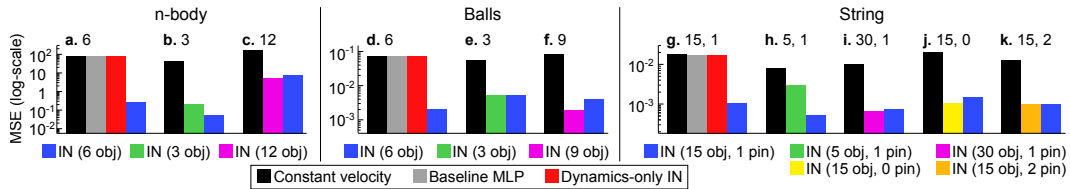

Figure 3: Prediction experiment accuracy and generalization. Each colored bar represents the MSE between a model's predicted velocity and the ground truth physics engine's (the y-axes are log-scaled). Sublots (**a-c**) show n-body performance, (**d-f**) show balls, and (**g-k**) show string. The leftmost subplots in each (**a, d, g**) for each domain compare the constant velocity model (black), baseline MLP (grey), dynamics-only IN (red), and full IN (blue). The other panels show the IN's generalization performance to different numbers and configurations of objects, as indicated by the subplot titles. For the string systems, the numbers correspond to: (the number of masses, how many ends were pinned).

Table 1: Prediction experiment MSEs

| Domain | Constant velocity | Baseline | Dynamics-only IN | IN |
|--------|-------------------|----------|------------------|-----|
| n-body | 82 | 79 | 76 | **0.25** |
| Balls | 0.074 | 0.072 | 0.074 | **0.0020** |
| String | 0.018 | 0.016 | 0.017 | **0.0011** |

(n-body MSE 19, string MSE 425). The IN presumably learns the gravitational and spring potential energy functions, applies them to the relations in their respective domains, and combines the results.

## 5  Discussion

We introduced interaction networks as a flexible and efficient model for explicit reasoning about objects and relations in complex systems. Our results provide surprisingly strong evidence of their ability to learn accurate physical simulations and generalize their training to novel systems with different numbers and configurations of objects and relations. They could also learn to infer abstract properties of physical systems, such as potential energy. The alternative models we tested performed much more poorly, with orders of magnitude greater error. Simulation over rich mental models is thought to be a crucial mechanism of how humans reason about physics and other complex domains [4, 12, 10], and Battaglia et al. [3] recently posited a simulation-based "intuitive physics engine" model to explain human physical scene understanding. Our interaction network implementation is the first learnable physics engine that can scale up to real-world problems, and is a promising template for new AI approaches to reasoning about other physical and mechanical systems, scene understanding, social perception, hierarchical planning, and analogical reasoning.

In the future, it will be important to develop techniques that allow interaction networks to handle very large systems with many interactions, such as by culling interaction computations that will have negligible effects. The interaction network may also serve as a powerful model for model-predictive control inputting active control signals as external effects – because it is differentiable, it naturally supports gradient-based planning. It will also be important to prepend a perceptual front-end that can infer from objects and relations raw observations, which can then be provided as input to an interaction network that can reason about the underlying structure of a scene. By adapting the interaction network into a recurrent neural network, even more accurate long-term predictions might be possible, though preliminary tests found little benefit beyond its already-strong performance. By modifying the interaction network to be a probabilistic generative model, it may also support probabilistic inference over unknown object properties and relations.

By combining three powerful tools from the modern machine learning toolkit – relational reasoning over structured knowledge, simulation, and deep learning – interaction networks offer flexible, accurate, and efficient learning and inference in challenging domains. Decomposing complex systems into objects and relations, and reasoning about them explicitly, provides for combinatorial generalization to novel contexts, one of the most important future challenges for AI, and a crucial step toward closing the gap between how humans and machines think.

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
