[Supplementary Material]

# 1 Supplementary material

## 1.1 Experimental details

### 1.1.1 Physics engine details

Every simulated trajectory contained the states of the objects in the system on each frame of a sequence of 1000 one-millisecond time steps. The parameters of each system were chosen so that a diverse set of dynamics could unfold within the trajectory. On each step, the physics engine took as input the current state of the system, calculated the forces associated with each inter-entity interaction, and applied them to the individual entities as accelerations by dividing by the entity's mass, parameterized as inverse-mass, $a = Fm^{-1}$ in both the engine and model input. This also allows static objects to be represented as having $m^{-1} = 0$. The previous positions and velocities, and newly computed accelerations, were input to an Euler integrator to update the current velocities and positions of the entities. By using a custom engine we were able to have perfect control over the details of the simulation, and use one engine for all physical domains. It produced trajectory rollouts that were indistinguishable from those of an off-the-shelf simulation engine, but was highly efficient because it could perform thousands of runs in parallel on a GPU, allowing millions of simulated steps to be generated each second.

### 1.1.2 Physical domains

**n-body** All objects in n-body systems exerted gravitational forces on each other, which were a function of the objects' pairwise distances and masses, giving rise to very complex dynamics that are highly sensitive to initial conditions. Across simulation runs, the objects' masses varied, while all other non-dynamic variables were held constant. The gravitational constant was set to ensure that objects could move across several hundred meters within the 1000 step rollout. The training scenes always included 6 objects, and for testing we used 3, 6, and 12 objects. The masses were uniformly sampled from $[0.02, 9]$ kg, their shapes were points, and their initial positions were randomly sampled from all angles, with a distance in $[10, 100]$ m. We included two classes of scenes. The first, orbit systems, had one object (the star), initialized at position $(0, 0)$, with zero velocity and a mass of 100 kg. The planets' velocities were initialized such that they would have stable orbits around the star, if not for their interactions with other planets. The second, non-orbit systems, sampled initial $x$- and $y$-velocity components from $[-3, 3]$ m/s. The objects would typically begin attracting, and gave rise to complex semi-periodic behavior.

An n-body system is a highly nonlinear (chaotic) dynamical system, which means they are highly sensitive to initial conditions and extended predictions of their states are not possible under even small perturbations. The relations between the objects corresponded to gravitational attraction. Between simulation runs, the masses of the objects were varied, while all other non-dynamic variables were held constant (e.g., gravitational constant) or were not meaningful

(e.g., object scales and shapes). The gravitational force from object $i$ to $j$ was computed as, $F_{ij} = \frac{Gm_i m_j (x_i - x_j)}{\|x_i - x_j\|^3}$, where $G$ is the gravitational constant. The denominator was clipped so that forces could not go too high as the distances between objects went to zero. All forces applied to each object were summed to yield the per-object total forces.

**Bouncing balls**  The bouncing balls domain still had all-to-all object relations–collisions–and any object could collide with any other, including the walls. But colliding objects are more difficult to simulate than the gravitational interactions in n-body systems, and our bouncing balls domain also included more variability in object attributes, such as shape, scale, and mass (as before), as well as relation attributes, such as the coefficient of restitution. The data distribution was much more challenging: for more than 99% of the time steps, a ball was not in contact with any others, and its next-step velocity equaled its current velocity. For the remaining steps, however, collisions caused next-step velocities that was a complex function of its state and the state of the object it collides with. The training scene contained 6 balls inside a box with 4 walls, and test scenes contained either 3, 6, or 9 balls. The balls' radii were sampled uniformly from $[0.1, 0.3]$ m, and masses from $[0.75, 1.25]$ kg. The walls were static rectangular boxes, positioned so that the horizontal and vertical lengths of area they enclosed varied independently between $[1, 3]$ m. The balls' initial positions were randomly sampled so that all balls fit within the box and did not interpenetrate any other object, with initial $x$- and $y$-velocity components sampled uniformly from $[-5, 5]$ m/s. The restitutions, an attribute of their collision relations, was sampled uniformly from $[0.4, 1]$.

Rigid body collision systems are highly nonlinear (chaotic) dynamical systems, as well. Collision forces were applied between objects when they began to interpenetrate, using two-step process: collision detection between all objects, and resolution of detected collisions. A detected collision between two objects meant that their shapes overlapped and their velocities were causing them to approach each other. To resolve a partially inelastic collision, the post-collision velocities of each object were computed, and forces appropriate to effect these velocities on the subsequent time step were then calculated and applied the the objects. This resulted in realistic bouncing ball trajectories.

**String**  Our string domain used multiple types of relations (springs and collisions), relation structures that were more sparse and specific than all-to-all, and variable external effects. Each scene contained one string, comprised of point masses connected by springs, and one static, rigid circle positioned below the string. Gravitational acceleration, varied across simulation runs, was applied so that the string always fell, usually colliding with the static object. Each training scene contained a string with 15 point masses, and test scenes contained either 5, 15, or 30 mass strings. In training, one of the point masses at the end of the string, chosen at random, was always held static, as if pinned to the wall, while the other masses were free to move. In the test conditions, we also included

Table 1: Prediction experiments - Generalization MSEs

| Model | n-body | | Balls | | String | | | |
|---|---|---|---|---|---|---|---|---|
| | 3 | 12 | 3 | 9 | 5, 1 | 30, 1 | 15, 0 | 15, 2 |
| Const. vel. | 45 | 166 | 0.054 | 0.084 | 0.0080 | 0.010 | 0.020 | 0.012 |
| IN (within) | 0.21 | **5.2** | **0.0053** | **0.0019** | 0.0030 | **0.00066** | **0.0010** | **0.00097** |
| IN (transfer) | **0.052** | 7.8 | **0.0053** | 0.004 | **0.00052** | 0.00074 | 0.0014 | **0.00097** |

strings that had both ends pinned, and no ends pinned, to generalization. The string's masses were sampled from $[0.05, 0.15]$ kg. The springs' rest lengths were set to 0.2 m, spring constants to 100, and damping factors to 0.001. The static object's $x$-position was sampled from $[-0.5, 0.5]$ m, $y$-position was sampled from $[-1, -0.5]$ m, and radius from $[0.2, 0.4]$ m. The string mass-rigid object coefficient of restitution was sampled from $[0, 1]$. The gravitational acceleration applied to all non-static objects was sampled from $[-30, -5]$ m/s$^2$.

The spring force from object $i$ to $j$ was computed using Hooke's law as, $F = C_s(1 - \frac{L}{\|x_i - x_j\|})(x_i - x_j)$, where $C_s$ is the spring constant and $L$ is the spring's rest length. A damping factor, which was proportional to the difference in the objects' velocities, was also applied. Collision forces between the string's masses and the static, rigid object were calculated as described in the bouncing balls section above.

## 1.2 Results details

The prediction experiment's generalization performance MSEs are shown Table 1.