[Reviews · NeurIPS 2016]

Reviewer 1

Summary

The paper proposes interaction networks, a model for learning interactions of objects in complex systems. The model is intended to make dynamical predictions on the system and infer abstract properties such as potential energy. The systems are represented as graphs and are input into a deep neural network that composes of a relation- and an object-centric stage. The network has been trained on simulations of three different complex domains of classical physics. The authors claim to have created the first general-purpose, learnable physics engine.

Qualitative Assessment

The paper is very well-written, experimentally and theoretically sound, and generally easy to comprehend and nicely self-contained. The proposed model is (as far as I can tell) genuinely novel and fulfills the exciting promise of a general-purpose, learnable physics engine with impressive results. I have only a few remarks, mostly concerning the paper's presentation: 1) In my regard, there are some issues with the introduction of the paper, which seems a bit unfocused. While the model is excellently described in its definition of page 3, the introduction appears too abstract in order to provide a foundation for the technical details and referencing Figure 1 before the model definition appears rather confusing. To fix this, a short general description of the data stream through the network might be helpful, as well as a short remark pointing towards section 2. This could also help to convey how the model is a combination of structured models, simulation and deep learning (line 25). 2) Some details and inaccuracies could be fixed in the Experiments section. - In line 202, you state that 'each object state "could" be further divided into a dynamic state component and a static attribute component'. Is this currently the case or a suggestion for future implementations? - In line 217, you state that the noise on the data is "annealed"; in epochs 50 to 250. By annealing, do you simply mean 'reduced'? The word suggests a more complex algorithm (as in simulated annealing). - From line 222, I inferred that supposedly static objects were not implemented as static by the IN. Is that correct? Either way, this could be clarified in the beginning of the 'Physical reasoning tasks' section. 3) As an added suggestion, possibly for future work, the current experiments leave room for some questions. While the highly non-linear dynamics of the physical systems are mentioned and the resulting limited testability of the model is thoroughly discussed, this issue could motivate some further investigations regarding the physical plausibility of the model's predictions. Beyond judging the trajectories on appearing visually realistic, one could check for conservation of the system's global physical properties (energy, momentum) or attempt to quantify the plausibility of the trajectories with respect to the system's complexity, possibly using non-linearity measures used in non-linear data analysis.

Confidence in this Review

2-Confident (read it all; understood it all reasonably well)


Reviewer 2

Summary

The authors propose "Interaction Networks", a neural network architecture that learns to propagate physical dynamics based on state variables (position, velocity), like a "learned" physics engine. The network architecture encodes a bunch of important physical priors (i.e. learning directly on state variables; assuming single-object and pairwise interactions only; assuming only a subset of interactions occur based on the object types, which are hardcoded in the input; also I think accumulating forces by sum, although it's a bit unclear). They train and evaluate interaction networks on a few different physical scenarios (n-body interactions, ball-and-string, colliding balls), and find that they beat some weak baselines. They also provide videos which show pretty convincing qualitative behavior.

Qualitative Assessment

There have been a number of papers recently that use neural networks to perform physics reasoning tasks. Most of these papers learn from visual input rather than state variables, in fact the previous work by Fragkiadaki et al. on billiard ball prediction (http://arxiv.org/pdf/1511.07404v3.pdf) is extremely similar to this work, except Fragdiadaki only test on billiard ball prediction, and their (also object-centric) model takes synthetic images rather than state variables as input. The authors have simplified the physics prediction problem with a bunch of assumptions (mentioned above). On one hand, that makes the problem a lot easier (1 million training examples seems way overkill for a problem with just 20-100 variables...) But these assumptions are interesting because they are a potential small set of priors that would allow their physics model to generalize in reasonable ways to data outside the training set, which is well-demonstrated by this work (which the authors should emphasize more). I think that the authors could improve this work if they compare more with prior work, and especially if they could show that their model generalizes in ways that e.g. the Fragkiadaki model does not, and/or run ablation studies to determine exactly which of their model's simplification/assumptions allow it to generalize in interesting ways. It would help if the authors provided an easier-to-follow text description of the model specified in Eq. 1 and 2, I found it hard to follow and figure out e.g. how the aggregation function works (what does "The aggregation function...collects all effects...merges them... to form a set of object model inputs"). Since the NN instantiation of the model is not described in depth either, I think it would be hard to replicate this work. Nit: Figure 2 caption, make clear that the model is trained on object positions, not images. Nit: Figure 3 is very confusing because each of the experiments has a different subset of conditions (e.g. in d,e,f each has different bar colors). Maybe rethink how to display this information. Overall, I would give this a borderline accept as a poster, although it could definitely benefit from some further work.

Confidence in this Review

3-Expert (read the paper in detail, know the area, quite certain of my opinion)


Reviewer 3

Summary

This paper presents an architecture for reasoning about complex physical interactions between objects, called an Interaction Network. The model in its general form takes in object states and interactions between objects as input and simulates the system forward to produce next object states or to reason about physical quantities. The paper proposes an implementation of the interaction network using deep neural networks, and discusses results on three simulated scenarios. Results are promising and show good generalization to varying object shapes, sizes and physical properties.

Qualitative Assessment

Technical Quality & Novelty: This paper combines ideas from structured models and NNs, to formulate the physics simulation problem as a multi-graph that can be learned from data. The key idea is in separating the computation into node-based and edge-based operations (and defining these operations in a specific manner), while imposing a single learned model for all interactions and objects. This allows the network to be structured, data-efficient and be somewhat invariant to the number/ordering of objects in the scene. I like this idea of structuring the computation based on the task as this adds priors which make the learning problem more tractable. The results show that the interaction network has good performance and generalization across different types of physics simulation problems. The generalization results validate some of the choices made in the model architecture and are very promising. On the other hand, I feel that imposing a single model for all the objects/interactions can restrict the representation power of the interaction network - for example when dealing with dynamics at different time scales. One way to get around it might be to learn an ensemble of models from which the network can choose a single or a weighted combination of effect(s). Nonetheless, the ideas presented are novel and the paper is technically good. Impact: This paper gets at an important problem - learning to predict physics and the effects of actions on the real world. The model proposed by this paper is general and can be applied to many problems in this context eg. robotics. One major concern I have regarding its applicability to real world problems (and from the paper as a whole) is the fact that the system needs the shape, mass of all the objects and the types/parameters of all the potential interactions. In the real world, this means solving a very hard system identification / detection problem for which no general solutions currently exist. I feel that this limits the applicability of the current work. There are significant areas for future research, potentially in real-world predictions using varying noisy input/output modalities (maybe images). A starting point towards this can be to test the performance of the system given noisy object attributes / relations. Clarity: The paper is well written and easy to understand. The authors could have aggregated the videos for each scenario/number of balls together. It is hard to compare the results from separate videos. Overall, this paper presents an interesting architecture for modeling physical interaction. It makes good contributions towards an important problem in ML/Robotics.

Confidence in this Review

3-Expert (read the paper in detail, know the area, quite certain of my opinion)


Reviewer 4

Summary

This paper aims at understanding some aspects of systems of objects interacting with each other. They focus on simulations from physical systems. In this context, they look at two problems: given a system of objects interacting with each other, can we a) predict the velocity of all the objects at the next time step, given their current state? b) predict more abstract quantities of the system, such as energy? The inputs to the model are the state of the objects at a given time, and information about binary relations between them ("senders" and "receivers"). The proposed method trains three neural networks, which do the following: 1) given two objects which are related, output the effect of this relation on the dynamics. 2) given an object, any external elements applied to it, and the effect of all objects for which this object is a receiver, compute the predicted next state of interest (for instance, velocity). 3) given the output of the second network, compute a quantity of interest about the system, such as energy. The network is trained with single step prediction, to predict the ground truth of the next time step. The networks can generalize to an arbitrary number of relations and objects at test time. The algorithm is tested on three simulated problems (n-body system, bouncing balls and strings). The authors report significant improvement in next step prediction with respect to defined baselines, and prove that the method is able to generate realistic system behaviors.

Qualitative Assessment

To my knowledge, the method proposed in the paper in novel, and the results show that the IN can model physical systems with complex interactions. The behaviors generated by IN are realistic and provide an exciting direction for further research. I have certain concerns, which I explain below. 1) The paper proposes a new algorithmic method, but lacks theoretical analysis of the concept or the results. There is extensive theoretical research in physical dynamical systems which could be linked to. In my opinion, adding such analysis could benefit the paper by helping the reader understand the high number of steps in the algorithm. It could also add interesting validation methods. One could for instance address a system for which the dynamics are well studied, and can be solved in closed form. A possible candidate would be a 2 body system. The authors could use such an example both to guide the reader through the necessary steps of the method, and also to provide experimental check against the ground truth, as well as further theoretical insight. 2) The presented method inputs to phiO the sum of the effects predicted by phiR. Why should we expect this additive structure to hold in practice? I am aware that the authors have considered the more realistic situation in which nth order relations are available, instead of binary ones (cf line 114). While this is certainly out of the scope of this work, isn't it the case that assuming this additive effect structure if too strong an assumption? 3) I would have appreciated to see the code for this paper, which would have clarified several questions regarding the implementation. Some minor comments: - How is the size of the effect layer chosen (here, 50)? - l 215: what are the "similar statistics" referred to? - l 180: aren't there more than n(n-1) relations, i.e. the walls need to be added? - l211: how is the data subsampled? Do you select pairs of points at random? - l35: typo, missing parenthesis. - l122: typo, can *be* trained - l234: typo, output*s* - When referring to a section, numbers would be helpful (ie line 136).

Confidence in this Review

2-Confident (read it all; understood it all reasonably well)


Reviewer 5

Summary

The authors aim to develop a learning-based physics engine. The engine developed in the paper take as input an interaction graph. This graph has nodes corresponding to objects and edges corresponding to relations between the objects. At each step of the simulation, a trained MLP is used to predict the effects of inter-object interactions. Then, a separate trained MLP is used to predict the state of the object at the next time-step taking into account both the previous state of the object and the effects of its relationship with other objects. The authors train and test the proposed model on three different types of scenarios: n-body systems; balls bouncing in a box; and strings composed of springs that collide with rigid objects. They demonstrate accurate predictions and generalization ability of the model one time-step into the future.

Qualitative Assessment

A. Experiments * It seems only one timestep into a future is explored (lines 252-253). The authors should have tried to evaluate the model over a longer timescale. Especially in the case of balls bouncing in a box, it seems much harder to correctly predict the future positions as errors in positions tend to accumulate over the tilmestep. * The qualitative results (Fig. 2) indicates the model inaccurately predicts over a longer timescale - a property rather important for a physics engine. B. Model * The presented model works under quite a strong assumptions where an 'interaction graph' is given. Works such as Fragkiadaki et al. [5] (ICLR'16) demonstrate similar abilities but under weaker modelling assumption (from a raw visual input). * It also seems the model requires quite a lot of training data (1 million). Hence it is hard to tell if the model has any possibility to generalize outside of the 'simulation' domain. C. Motivation * It is unclear for me what is the main motivation/purpose behind the submission. If simulation and 'interaction graph' are available then arguably we can use 'laws of physics' without a need to learn anything. Although other works (e.g. [5]) still use simulation, the interactions are learnt from the data only, and hence such a line of research ([5]) may have an impact for real-world scenarios. Likewise, in real-world scenarios extracting an 'interaction graph' may become a quite difficult problem -- and such graph is likely to be 'corrupted by noise'. Such issues are not addressed in the submission. --- I liked the rebuttal, and modified my score a bit. However, there are things that still bother me. Regarding the experiments: The submission lacks of quantitative results on long-term predictions (either with MSE or by proposing an alternative measure). The MSE results given in Table 1 (and majority results in the supplementary material) already exhibit small errors. Hypothetical future work that builds upon the submission would arguable explore longer-term predictions, and need a comparison. Moreover, qualitative results may be good in the 'intuitive physics' domain, but to build 'physics engine' or to allow reproducibility/comparisons automatic measures showing deviations from the ground truth are also important. I also think the Table from supp. should be moved to the main paper -- it shows quite important results. It would also be great if the authors make the dataset / code publicly available. Regarding the motivation/impact: I agree with the authors that predicting functional forms in the interaction networks might be useful in many scenarios ranging from intuitive physics, learning engines of systems with unknown simulation, or social networks, but none of the aforementioned tasks have been explored in the submission. For instance, in the 'intuitive physics from images' domain, the authors have to extract a noisy graph representation of an image. Is IN competitive with the work Fragkiadaki et al. in this scenario? I see the experiments shown in the submission as valuable additional experiments done in a more controlled environment but they miss real applications. Therefore, I feel the work is somehow incomplete, but I also agree it has a good potential, and is a nice initial study in a controlled environment under the assumption that the graph is given.

Confidence in this Review

2-Confident (read it all; understood it all reasonably well)